# Application of Advanced Non-Linear Spectral Decomposition and Regression Methods for Spectroscopic Analysis of Targeted and Non-Targeted Irradiation Effects in an *In-Vitro* Model

**DOI:** 10.3390/ijms232112986

**Published:** 2022-10-26

**Authors:** Ciara Slattery, Khanh Nguyen, Laura Shields, Isabel Vega-Carrascal, Sean Singleton, Fiona M. Lyng, Brendan McClean, Aidan D. Meade

**Affiliations:** 1School of Physics and Clinical and Optometric Sciences, Technological University Dublin, City Campus, Dublin 8, Ireland; 2Radiation and Environmental Science Centre, Focas Research Institute, Technological University Dublin, Camden Row, Dublin 8, Ireland; 3Medical Physics Department, St. Luke’s Hospital, Highfield Road, Rathgar, Dublin 6, Ireland

**Keywords:** Fourier transform infrared microspectroscopy (FTIRM), out of field effects, t-stochastic neighbourhood embedding (t-SNE), principal components analysis (PCA), support vector machine (SVM), extreme gradient boosting regression (XGBR)

## Abstract

Irradiation of the tumour site during treatment for cancer with external-beam ionising radiation results in a complex and dynamic series of effects in both the tumour itself and the normal tissue which surrounds it. The development of a spectral model of the effect of each exposure and interaction mode between these tissues would enable label free assessment of the effect of radiotherapeutic treatment in practice. In this study Fourier transform Infrared microspectroscopic imaging was employed to analyse an *in-vitro* model of radiotherapeutic treatment for prostate cancer, in which a normal cell line (PNT1A) was exposed to low-dose X-ray radiation from the scattered treatment beam, and also to irradiated cell culture medium (ICCM) from a cancer cell line exposed to a treatment relevant dose (2 Gy). Various exposure modes were studied and reference was made to previously acquired data on cellular survival and DNA double strand break damage. Spectral analysis with manifold methods, linear spectral fitting, non-linear classification and non-linear regression approaches were found to accurately segregate spectra on irradiation type and provide a comprehensive set of spectral markers which differentiate on irradiation mode and cell fate. The study demonstrates that high dose irradiation, low-dose scatter irradiation and radiation-induced bystander exposure (RIBE) signalling each produce differential effects on the cell which are observable through spectroscopic analysis.

## 1. Introduction

Prostate cancer is the second most common cancer amongst men and the fifth leading cause of death worldwide due to it being predominantly asymptomatic in early stages, with a higher prevalence in developed countries [1]. As of 2018 the age-standardised incidence rate of the disease was 75.8 per 100,000 in Western Europe with a mortality of 10.1 per 100,000 [2].

Treatment is not always necessary for this disease, as in the early stages it progresses slowly. The exact course of treatment prescribed is dependent on age and overall health. Should treatment be pursued, the prostate may be surgically removed or the disease treated with radiotherapy alone, or via radiotherapy accompanied with hormone therapy [3]. External beam radiation (EBR) is the most common type of radiation therapy used for treatment of prostate cancer.

During treatment of solid cancers with EBR, cells outside the treatment volume may experience damage induced by radiation induced bystander effects (RIBE) [4]. Damage may also occur to surrounding healthy tissues outside the treatment field due to transmission and scattering of radiation in conjunction with RIBE from the irradiated tumour to the surrounding normal tissue [5]. Radiation induced bystander effects are currently understood to involve cell to cell signalling through gap-junctional intercellular communication and the release of soluble factors into the intercellular space, with downstream effects on DNA damage and mutation, cell survival and cell death in cells exposed to bystander signals [6,7]. The molecular species involved in the response include calcium, chemokines, cytokines, and reactive oxygen species together with exosomes [6]. All of these factors induce stress responses in exposed cells which are similar in action to inflammatory responses and are differentially expressed in cancer and normal tissue, with normal tissue being more sensitive to bystander responses overall [6]. It is also known that in terms of dose response the effect saturates at 1 Gy, although bystander responses are seen in unexposed cells exposed to factors from cells irradiated at both low and high doses [6]. Ultimately these effects do have a role in modulating the response to irradiation at the organism level (termed ‘abscopal responses’) through the release of ‘clastogenic factors’ which include those mentioned above, and are therefore relevant for cancer therapy [6,7].

The propensity for cells to survive out-of-field irradiation is dependent on their intercellular communication. Cellular radiosensitivity is also an important factor in the response to out-of-field effects. Previously it has been demonstrated that out-of-field (OF) irradiation has the potential to have a detrimental effect on the proliferation of normal prostate cells positioned outside the primary radiation beam when a clinically relevant dose of 2 Gy is delivered in-field (IF) [5].

In Fourier transform infrared (FTIR) spectroscopy the complete biochemical fingerprint of a biological sample is acquired, where the natural frequency of vibration of the bonds within the molecules of a sample are excited without extraneous labelling. In biological or clinical applications, the technique has been shown to provide highly sensitive phenotyping of the sample [8,9,10], and has demonstrated many applications in radiobiological and radiation science over the past 15 years. In early studies by Matthews et al., Raman spectral signatures of breast [11], lung [11] and prostate [11,12] were analysed post exposure to 2–50 Gy X-ray photons. Since then, studies have demonstrated a dose dependence to spectral features in X-ray irradiated human mammary epithelial [13], human neuroblastoma cells [14] and human prostate cancer cells [15], which aligns with research which has demonstrated the potential of both Raman and FTIR spectroscopy to provide the opportunity for spectral radiation dosimetry in rapid response applications [16,17]. Importantly for the current research previous studies have also demonstrated the detection of radiation induced bystander effects from exposure of keratinocytes to gamma ray photons [18] and from exposure of prostate cancer cells to protons at 1MeV and 2MeV [19]. In both cases a degree of characterisation of the bystander response was achieved using the FTIR spectra of bystander irradiated cells, which the present study moves forward.

Pre-clinical studies have also taken place on the spectral alterations seen in non-squamous cell lung [20] and breast [21] cancer xenografts irradiated with 5 and 15 Gy doses from a 6MV photon beam. Additional studies with the NSCLC model revealed that a time course in Raman signatures could be observed which were referenced to signatures of hypoxia and reoxygenation [21]. Further investigations have observed the relationship between Raman spectral features and those of the DNA DSB damage response and the capability to classify radiosensitivity in ex vivo cultured lymphocytes [22,23]. Latterly Raman spectroscopy has been applied to the detection of radioresistance in an *in-vitro* isogenic oesophageal cancer model [24]. Key here was the demonstration that a signature of radioresistance could be confined to a single Raman band (centred at 977 cm^−1^), rather than alterations across the spectrum as seen in other works. This alteration is due to a biochemical species which remains unidentified.

Further recent research has also demonstrated that these technologies can be applied to the detection and monitoring of radiobiological responses in the clinic, in particular when applied to liquid biopsies from patients undergoing external beam radiotherapeutic treatment for prostate cancer [25,26]. In this particular context it has been well demonstrated that in external beam radiotherapy a complex interplay between lethal and sublethal damage responses occurs in the directly irradiated tissue, together with sublethal low dose effects owing to scatter and molecular signaling [4,27,28,29,30]. Previous work has demonstrated the potential for Raman and IR spectroscopy to identify both radiobiological effects at high doses and low dose non-targeted effects [18,19]. In the present study, the potential for FTIR spectroscopy to identify and quantify mixed lethal and sublethal radiobiological damage in an *in-vitro* prostate cancer and normal tissue model was studied.

In this study Fourier transform infrared (FTIR) microspectroscopy (FTIRM) was used to analyse these effects in a prostate model *in vitro*, for which radiobiological results have previously been reported elsewhere [5]. Spectral analysis of an experiment in which a simulation of intercellular communication between irradiated prostate tumour cells (LNCaP line) and cells of the surrounding normal prostate tissue (PNT1A line) is conducted, where communication between the cells is both prohibited and permitted. In the experiment conducted by Shields et al. [5]. LNCaP cell lines were irradiated with X-ray doses of 2 Gy (labelled as LNCaP IF (in-field) in this instance). The PNT1A cells were then exposed to the medium from the LNCaP cells with or without a prior exposure to a low dose of ionising radiation (0.2 Gy) which simulates the effect of direct exposure of the normal tissue adjacent to tumour in prostate cancer treatment.

The treatment classes therefore allow the comparison of the effect of various exposure modes to normal tissue (PNT1A cell line). These include (i) direct irradiation at low doses (labelled as PNT1A OF), (ii) direct irradiation plus exposure to secreted factors from the tumour (labelled as PNT1A OF ICCM) and (iii) exposure to secreted factors from the tumour (labelled as PNT1A 0 Gy ICCM), each of which can be compared to the sham-irradiated control sample (labelled PNT1A 0 Gy). These exposure modes therefore allow an intercomparison of exposure where intercellular communication via secreted factors from the tumour to surrounding normal tissue does not occur (PNT1A OF) to where it does occur when the normal cells adjacent to the tumour lie within the scatter from the treatment beam and receive both a direct exposure at low doses and exposure to secreted factors (PNT1A OF ICCM).

In this study, samples prepared in parallel to those used by Shields et al. [5] in the primary study are used for spectroscopic analysis and modelling to provide the opportunity to identify spectral features which differentiate on exposure mode. A number of differentiating spectral features are highlighted which may be associated with the different exposure modes and may offer direction to future investigation in this area.

## 2. Results

### 2.1. Visualisation of Spectra

Figure 1 displays the mean spectra for each of the cell lines by treatment type. Very slight spectral differences were observed with few distinguishing features to the naked eye.

Visualisation of the principal component scores was conducted with a separate decomposition applied to spectra from the PNT1A line and the LNCaP line, together with each of their treated spectra (Figure 2). Here, again it is not possible to visually separate each of the cell lines, nor their treatment classes using this approach.

t-SNE decomposition of the spectral data was employed to cluster and visualise the spectral data such that inter-cluster relationships might reveal spectral encoding of biochemical alterations from each of the irradiation modes. In generating these plots the perplexity, learning rate and number of iterations were varied and the t-SNE decomposition was applied separately to the PNT1A spectra and LNCaP spectra, incorporating all irradiation modes for each. We have also observed that spectral processing is key to obtaining interpretable t-SNE visualisations with FTIR data, as t-SNE appears to be quite sensitive to the choice of processing employed. Here, we have utilised 2nd order derivatised data which are subsequently standardised to have mean of zero and standard deviation of 1 using the standard normal variate transformation. A set of representative t-SNE scores plots from this analysis are provided in Figure 3a,b.

### 2.2. PCA-SVM Modelling

For classification of spectra PCA-SVC was employed, where a brute-force grid-search approach with three-fold stratified cross-validation was utilised to choose the optimal hyperparameters yielding optimal classification of each spectral class. Data was randomly separated into a training set for model optimisation and a testing set in a 70:30 split. A pipeline was created which iterated through the data by principal component and by hyperparameter option to identify the preferred hyperparameters for classification at cross-validation. Optimised models were then applied to the test set. This approach was applied separately to PNT1A spectra and LNCaP spectra, with F_1_-score used as the metric of classification performance, and models were generated for 20 separate data randomisation epochs.

It was found that the radial basis function kernel with a cost parameter (C) of 1 and a gamma parameter which was set as 1*/(number of features × variance of data)* yielded optimal classification models with a number of principal components in the region of 10, as depicted in Figure 4a,b. On further examination a feature selection optimisation found that models utilising scores for principal components from 1 to 5 yielded F_1_ scores at cross-validation in excess of 0.97 at both training and testing. Figure 5 depicts the loadings to principal components 1 to 3.

### 2.3. CLS Spectral Fitting

The results of the CLS fitting of the each individual cell class are depicted in Figure 6a,b. The broad agreement between the CLS fit and the mean spectra for each class is excellent with the deviation in the case of the PNT1A spectrum in the region of 0.2% and the deviation in the case of the LNCaP spectrum in the region of 0.5%.

The statistical significance between the CLS regression coefficients was assessed using Welch’s ANOVA [31] with the Games-Howell approach [32] used for post hoc pair-wise comparison between the distribution of coefficients by molecule and treatment class. The results of these tests are reported in Table 1, Table 2, Table 3 and Table 4, with non-significant *p*-values highlighted in red. The threshold for significance adopted here was *p* < 0.001.

### 2.4. XGBoost Regression against Cell Volume

XGBoost regression utilised the data in Table 5 as the target variable, which is taken from Shields et al. [5]. After the implementation of grid-search cross validation it was found that the optimal XGBoost regression model for IR spectra of the PNT1A cell line against cell volume utilised a number of individual trees in the regression ensemble 5 and a maximal tree depth of 5. The optimised model randomly sampled 50% of input spectra and 10% of the spectral features, with a learning rate (eta) of 0.01. An example of the association between the IR spectral data and cell volume is shown in Figure 7a, with the features selected by the algorithm across all of the independent epochs shown in Figure 7b.

## 3. Discussion

Beginning with the visualisation of spectra using graphing of t-SNE scores, for each of the experimental examples studied here the t-SNE scores plots deliver a set of interesting messages. In the case of the LNCaP cell line a clear differentiation between the spectra of the control and directly irradiated sample is provided. However, in the case of the PNT1A cell line the clustering is less pronounced which is suggestive of a number of features. In particular it is clear that there is strong spectral heterogeneity both within and outside each spectral class, suggesting strong variability in response to indirect or untargeted irradiation modes. While there is some separation of spectra from the control class from each of the other classes there is no distinct order amongst the spectra of the out-of-field irradiated and bystander irradiated classes. It is therefore not possible to draw conclusions on the basis of relative cluster position regarding the relative effect of each exposure mode on the PNT1A cell spectra. However, subclasses of cell spectra do appear to emerge from the bystander irradiated control, out of field irradiated and out of field irradiated sample which may be associated with the stage in the cell cycle at which cells were exposed.

The high classification rates observed using PCA-SVM with the PNT1A cell line and LNCaP lines suggest that the PC loadings may offer a means to identify spectral features which discriminate between the treatment classes and between targeted and non-targeted irradiation effects. For interpretation purposes we have depicted the loadings to principal components 1 to 3 in Figure 5 for the sake of brevity. The loadings to these PCs account for 68% of the variance in the case of the PNT1A cell line and 87% in the case of the LNCaP cell line. In the interpretation of the loadings here vibrational assignments listed in our earlier work are utilised [33].

In the case of the PNT1A cell line loadings to vibrational modes of nucleic acid (ν C-O in the region of 1044 cm^−1^ and ν C-O of RNA in the region of 1112 cm^−1^; ν as -C=O DNA, RNA in the region of 1696 and 1716), carbohydrate (νC-O in the region of 1202 cm^−1^; O-H deformation in the region of 1220 cm^−1^; ν O-H in the region from 3300 to 3488 cm^−1^), and protein (Amide II in the region of 1532 cm^−1^; Amide I 1654 cm^−1^) predominate, with some loadings to vibrations of methyl terminals (−CH_2_ and −CH_3_ νs and νas from 2840 to 2968 cm^−1^).

In the case of the PNT1A cell line the positions of the loadings in LV2 to the Amide II β-sheet (in the region of 1524 cm^−1^) and Amide I vibrations (1626 cm^−1^ and 1658 cm^−1^) appear to have shifted downwards by between 4 cm^−1^ and 6 cm^−1^. Likewise the positions of the vibrations of methyl terminals in LV1 (2840 cm^−1^ to 2968 cm^−1^), LV2 (2956 cm^−1^) and LV3 (2846 cm^−1^ to 2968 cm^−1^) are shifted by between 4 cm^−1^ and 10 cm^−1^, again towards lower vibrational frequencies. Each of these features suggests a weakening of structural strength within protein and lipid associated with cell signalling and cell death mechanisms owing to low dose irradiation.

The picture is similar for the loadings in respect of the direct irradiation of the LNCaP cell line, with loadings to the stretching vibrations of C-O, and O-H bonds of both nucleic acid and carbohydrate evident, together with loadings to Amide I, Amide II and Amide A of protein, and stretching vibrations of methyl terminals of both protein and lipid. An interesting feature in this instance is that the loadings, in general, remain at their normal positions with the exception of those for both the νs vibrations of −CH_2_ (in the region of 2856 cm^−1^) and νs, νas vibrations of −CH_3_ (at 2932 cm^−1^ and 2964 cm^−1^). In the latter instance these vibrational modes appear to be shifted towards higher vibrational frequencies by between 4 cm^−1^ and 12 cm^−1^, which is generally associated with the strengthening of the vibration. This suggests a spectral signature which potentially differentiates cells on exposure mode, with a weakening of the vibrational strength of protein and lipid seen in response to indirect (bystander) irradiation with a strengthening of the vibrations of methyl moieties in protein and lipid in response to direct irradiation. While this is an interesting feature which has not previously been observed it is difficult, in the absence of parallel biochemical assays, to ascribe any potential biological origin for this signal, and this remains an opportunity for further investigation.

Despite the existence of these interesting spectral features they do not provide clarity in relation to the spectral changes which differentiate on mode of indirect irradiation or direct irradiation. In this case, CLS analysis can play a role by highlighting molecular spectral species which are associated with each exposure mode. Within the results of CLS analysis it is apparent that that statistically significant differences in CLS fitting coefficients were observed between the majority of the treatment classes for the PNT1A and LNCaP cell lines against the control sample.

As a reminder of the structure of the reference primary experiment, for the PNT1A cell line several exposure modes were investigated; these are (i) a direct irradiation to 0.2 Gy (PNT1A OF), (ii) direct irradiation to 0.2 Gy together with exposure to intercellular molecular communication from the tumour cells (PNT1A OF ICCM) and (iii) exposure to intercellular molecular communication from the tumour cells (PNT1A 0 Gy ICCM). A reference sham-irradiated control sample (PNT1A 0 Gy) was also prepared. In the original experiment Shields et al. noted a number of important reference results:A reduction in colony volume between the PNT1A cells exposed to 0.2 Gy (PNT1A OF) and in unirradiated cells exposed to secreted factors from the tumour cells (PNT1A 0 Gy ICCM) when compared to cells irradiated with a low dose prior to exposure to secreted factors from the tumour (PNT1A OF ICCM);An increase in DNA double strand break (DSB) damage foci (γH2AX fluorescence measured via confocal microscopy) for all exposure modes, with the exposure of PNT1A cells to both a dose of 0.2 Gy and secreted factors from the tumour cells (PNT1A OF ICCM) producing a statistically significant increase in damage relative to the other exposure modes.

As per the conclusions of Shields et al. [5] we suspect that the increase in colony volume (and increase in cell survival observed by Shields) in the PNT1A OF ICCM cells is a signature of the adaptive response, whereby PNT1A cells have adapted their response to exposure to RIBE factors through a priming out-of-field dose, as observed in previous work [34]. In the PNT1A OF ICCM sample the CLS results in Figure 6a(ii) depict an increase in the spectral contribution of actin (protein) but a reduction in that associated with RNA. These results are broadly in alignment with the results of Shields et al. which observed an increase in DNA damage in this sample, which would be associated with a reduction in transcription during DNA repair. Coupled with this is the observation of a substantial decrease in the signal from cytochrome C which is a sensor of radiation-induced bystander exposure (RIBE) and is released from the mitochondria of cells undergoing radiation-induced apoptosis [35,36,37]. The loss of cytochrome C signal in this sample suggests a reduction in apoptosis which agrees with the increase in colony volume observed by Shields et al. Further evidence for this effect is seen in terms of the lack of change in the signal from lipid (phosphatidylcholine and phosphatidylinositol) and the reduction in the signal from vitamin C and E, both of which are antioxidants which scavenge reactive oxygen species [29]. Overall the spectral data supports the view expressed in Shields et al. that direct irradiation followed by exposure to secreted factors from cells directly irradiated to high doses nearby results in an adaptive response with improved sensing of DNA damage and repair, leading to an increase in cell volume which is associated with cellular proliferation. These effects are summarised within Table 6.

In the PNT1A cells directly irradiated with a 0.2 Gy dose (PNT1A OF) a reduction in cell volume and an increase in DNA DSB damage foci relative to the sham-irradiated control was seen by Shields et al. In Figure 6a(ii) the CLS fitting results a reduction in actin (protein), coupled with an elevation in glycogen, TGF-β, cytochrome C and IL8 was observed. Additionally, an elevation of the signal from phosphatidylinositol (a membrane lipid more commonly directed towards the cytosol) coupled with a reduction in the signal from phosphatidylcholine suggest that these features signify an increase in radiation induced apoptosis (protein coagulation and membrane blebbing), aligning with the observations of Shields et al. Similarly, in the PNT1A control cells exposed to secreted factors from directly irradiated cells (PNT1A 0 Gy ICCM) a reduction in cell survival was seen, though through secreted bystander factors as opposed to direct irradiation, and no increase in DNA damage was observed. Here, we see no change to the signal from actin (protein) or cytochrome C but a depletion of RNA and TGF-β, an increase in signal from phosphatidylcholine coupled to a decrease in that from phosphatidylinositol, and a reduction in signal from TGF-β, glycogen and vitamin E. Interestingly we also see an increase in the signal from IL8. It is known that TGF-β is a mediator of ROS signalling and DNA damage [38,39], while IL8 is a downstream product of the NF-κB pathway in directly and indirectly irradiated cells [37,40,41]. It may be the case that these observations lead to a picture of bystander signalling in this cell line which is not primarily ROS driven, but rather driven by cytokine elevation associated with extracellular signalling. These features are also summarised within Table 6.

Finally, Figure 6b(ii) depicts the results of CLS fitting analysis on LNCaP cells which were directly irradiated with high doses (LNCaP IF–irradiated with a dose of 2 Gy) versus their sham irradiated control. Here, we see a depletion of the signal from actin (protein), RNA and IL8 with a significant increase in signal from lipid (phosphatidylcholine and phosphatidylinositol), antioxidants (vitamin C and vitamin D) and cytochrome C. In the work of Shields et al. a significant increase in DNA DSB damage was seen in the directly irradiated LNCaP cells, although cell survival was not measured. Taken with the results seen within the PNT1A cell line these observations allow a picture to be constructed of the spectral changes occurring with differential exposure mode, which are summarized in Table 6. The overall picture emerging from the CLS analysis in Figure 6 and summarised in Table 6 is the complexity of the molecular events connecting to cell fate by exposure mode. It is clear that the cellular spectral biomarkers of each exposure mode are each distinct, though their association with observed DNA damage and ultimate cell fate are intricate, and are beyond the potential of the CLS analysis presented here. It must be noted that some of these changes may be cell line dependent and therefore may not be seen consistently across all experiments.

While the results of t-SNE, PCA-SVM and CLS analysis provide insights into the spectral features which originate in differential molecular effects that are associated with various modes of exposure of PNT1A cells, the identification of spectral features associated with cell fate (i.e., proliferation and survival) across exposure modes requires alternative approaches. The results of XGBoost regression of cell spectra against cell volume (representing a proxy for cell survival) are shown in Figure 7. Within Figure 7b the most frequently selected features are those that lie in the region from 1134 cm^−1^ to 1152 cm^−1^ which are generally associated with the ν C-O vibrations in carbohydrate. The remainder of the features which are most commonly selected by the models across each independent epoch are those in the region from 998 to 1016 cm^−1^ which are again associated with stretching of the C-O moiety in carbohydrate, and those from 3320 to 3568 cm^−1^ which are associated with the stretching vibration in the O-H groups of carbohydrate. Taken in their totality this suggests that the spectral features associated with carbohydrate moieties such as C-O and O-H may be considered as a spectral marker of cell proliferation as measured by cell volume.

This study has, for the first time, analysed spectral changes which are seen in an *in-vitro* model of prostate cancer treatment, where direct exposure of both the tumour and normal tissue which surrounds it results in complex interactions which are both intercellularly and extracellularly mediated and can both positively and adversely impact upon cell survival. Our results indicate that these effects can be differentiated in terms of the mode of action, i.e., whether the irradiation is at high or low doses, and whether there is a low dose exposure together with an exposure to extracellularly generated secreted factors. It must be acknowledged that no single spectral marker is available which differentiates each mode of action of the exposure, but rather there are a series of spectral responses which may characterise the mode of action. Of course the use of a 2D model employed here does not encapsulate the increased complexity which would be observed in out-of-field abscopal effects observed ex vivo or in vivo, despite the congruence between some bystander effects observed *in vitro* and in vivo [42,43]. In particular the lack of a complete immune response in our model system is a significant limitation which can only be overcome by the spectroscopic analysis of animal models and in vivo human samples [7]. However, the observations in this paper may allow further development of spectral models differentiating radiobiological responses in normal tissue and irradiated tumour towards clinical application.

## 4. Materials and Methods

### 4.1. Sample Preparation and Characteristics

#### 4.1.1. Cell Culture, Irradiation, Exposure to Irradiated Cell Culture Medium (ICCM) and Parallel Biological Assays

Samples were prepared as part of a previous study in which DNA damage and cell survival were interrogated in prostate cancer (LNCaP) and normal prostate (PNT1A) cell lines irradiated *in vitro* [5]. The following briefly describes the cell culture, treatment, irradiation and parallel biological analyses conducted in that study.

Both the LNCaP and PNT1A cell lines were cultured using RPMI media (Sigma-Aldrich^®^, Wicklow, Ireland) which contained 10% foetal bovine serum (Sigma-Aldrich). The cells were maintained in an incubator at 37 °C with the humidity set at 95% and 5% CO_2_ and were transferred to T-25 culture flasks (Sarstedt Ltd., Wexford, Ireland) for irradiation. Three technical replicates of each of the cell lines were treated with each irradiation and experimental treatment as outlined here.

Both LNCaP and PNT1A cells were irradiated simultaneously with the LNCaP cells irradiated in-field and the PNT1A cells kept at least 1cm from the field edge to simulate out-of-field (OF) scatter effects to normal cells. A total of 2 Gy was delivered to the LNCaP cell samples in-field (IF), with parallel LNCaP and PNT1A cell samples sham-irradiated (0 Gy). The dose delivered was validated using Gafchromic EBT3 film (Ashland Inc., Bridgewater, NJ, USA).

Subsequent to the irradiation, irradiated cell culture medium (ICCM) was harvested from irradiated LNCaP cells one hour after irradiation and used to expose both a sham-irradiated and out-of field irradiated PNT1A sample. This approach was used to simulate intercellular communication from irradiated prostate lesions to surrounding normal tissue. ICCM was harvested at this time point as this is standard procedure in our laboratory for preparation of ICCM. Early work by Mothersill and Seymour (1997) investigated the effect of time post irradiation of medium transfer and showed no significant difference in clonogenic survival between 1 and 60 h post irradiation [44].

Reference radiobiological measurements included measurements of clonogenic survival and DNA double strand breaks (as γH2AX mean fold increase and number of γH2AX foci). Here, clonogenic survival was measured using a GelCount^TM^ automated colony counter (Oxford Optronics, Oxford, UK), with measurements of cell volume acquired using this instrument.

#### 4.1.2. FTIR Spectral Acquisition and Pre-Processing

Cells of each sample were deposited onto calcium fluoride slides using methodologies detailed elsewhere [26]. Spectra were acquired in transmission mode using a Perkin Elmer Spotlight 400 imaging spectrometer within which the detector (mercury cadmium telluride) was cooled with liquid nitrogen. Spectral images were acquired using a 6.25 μm × 6.25 μm pixel size and a spectral resolution of 4 cm^−1^ over the range 720–4000 cm^−1^ with 128 scans per pixel. All data pre-processing steps were implemented in Python (v 3.9.12) using the OCTAVVS library for pre-processing [45]. Firstly, individual spectra were extracted from the images with outliers removed using Rosner’s test applied to the PC scores of spectra within a given class. The number of spectra in each class was down-sampled by a factor of 2 to improve computational efficiency. Subsequently spectra were corrected for atmospheric contributions and then scattering effects using the resonant-Mie scattering correction [46]. Spectra were smoothed using a Savitsky-Golay algorithm with an order of 5 and window of 15 points. Spectra were then truncated to the 800 cm^−1^ to 4000 cm^−1^ region. Finally, any residual baseline was removed using a concave rubber band algorithm and spectra were vector normalised. In totality these procedures resulted in a dataset of ~12,000 spectra for analysis.

### 4.2. Chemometrics and Machine Learning

All chemometric and machine learning approaches used in the study were written in Python 3.9.12 with scikit-learn (v. 0.21.3). All data visualisation utilised the matplotlib (v. 3.5.1) and seaborn (v. 0.11.2) packages. Statistical analysis utilised the pingouin package (v. 0.5.2) [47]. All classification algorithms utilised the F_1_ score as the metric of performance [48]. The following sections detail each of the methodologies which were employed.

#### 4.2.1. Principal Components Analysis

Principal components analysis (PCA) is a widely used dimensionality reduction method for large datasets. In the context of the current work PCA should be considered a dimensionality reduction technique which preserves the *global* structure of the data matrix, *X*, through mapping of the data from a high-dimensional space to a lower dimensional space via removal of the covariance in the dataset, while preserving as much ‘variability’ (i.e., statistical information) as possible [49].
(1)$$X=T.S.P^{T}$$

The lower dimensional space formed by the new variables or *principal components matrix*, *P^T^*, removes the covariance between variables in the original data space with the reduction being performed by solving an eigenvalue-eigenvector problem on the covariance matrix of the data in its original space [26] as per Equation (1) (where *T* is the matrix of principal components scores and *S* is the diagonal eigenvalue matrix. Within biophotonics PCA has become one of the first options for recourse for researchers as the principal components can be readily interpreted via providing loadings which may be related to the original spectral variables.

#### 4.2.2. Support Vector Machine

A support vector machine (SVM) is a statistical learning algorithm that is applied to supervised machine learning [50]. These methods are used for classification, regression, and outlier detection [51]. It is a non-parametric approach which is characterized by an efficient hyperplane searching technique which identifies the optimal separating hyperplane between the classes [52], where the hyperplane itself is located at the position of the support vectors which lie on the edge of the class distributions [51,52]. If we let (*x*_1_, *y*_1_), (*x*_2_, *y*_2_),…, (*x_m_*, *y_m_*), are training data vectors *x* with class labels *y* where *x_i_* ∈ *R^d^* denotes vectors in a *d*-dimensional feature space and *y_i_* ∈ {−1, +1} is a class label [52], the SVM in its linear form finds the optimal separating margin by solving the following optimization task:

Minimize $\left\{\frac{1}{2}{\left|w\right|}^{2}+C\sum_{i=1}^{m}\varepsilon_{i}\right\},\;\varepsilon_{i}\geq 0$

Subject to
(2)$$y_{i}\left(w^{T}x_{i}+b\right)\geq 1-\varepsilon_{i},\;\;\;i=1,2,\ldots .m$$
where *C* is a penalty value, $\varepsilon_{i}$ are positive slack variables, *w* is a vector normal to the separating hyperplane, and *b* is a scalar quantity.

If linear separation is not possible, it can be combined with a ‘kernel’ trick that implements a non-linear mapping to a feature space in which the linear separating hyperplane is identified [52]. The kernel technique enables higher dimensional, non-linear models to be developed [52], and is computationally efficient for datasets with high dimensionality through the use of a kernel function, $K\left(x,z\right)=\langle \phi \left(x\right)\cdot \phi \left(z\right)\rangle$ [53], which computes the separating hyperplane without carrying out a mapping to feature space [54]. Commonly used kernels include the radial basis function and polynomial kernels.

##### 4.2.3. t-SNE

t-SNE (t-distributed Stochastic Neighbour Embedding) is a novel nonlinear dimensionality reduction algorithm [55] which, unlike PCA which preserves the global structure of the original data, maintains the local structure of the data within the new ‘embedding’ space.

The method in which t-SNE operates is by converting the high-dimensional Euclidean distance between data points, *x_i_* and *x_j_* in a Cartesian coordinate system, into a conditional probability $p_{i|j\;}$. The probability density distribution of the neighbouring data points to *x_i_* are assumed as a Gaussian function centered at *x_i_* with a variance $\sigma_{i}$ such that the probability of *x_j_* to be selected as the neighbour of *x_i_* is given as [55]:(3)$$p_{j|i\;}=\frac{exp\left(-\frac{\|x_{i}-x_{j}{\|}^{2}}{2\sigma_{i}^{2}}\right)}{\sum_{k\neq i}\exp\left(-\frac{\|x_{i}-x_{k}{\|}^{2}}{2\sigma_{i}^{2}}\right)}$$

However, $p_{j|i}$ is not equal to $p_{i|j}$, which causes SNE to be predisposed to outliers, such as $x_{i}$ having a very small value for $p_{j|i}$ which causes its embedded location to become irrelevant. The similarity of data points $x_{i}$ and $x_{j}$ are calculated as the joint probability [55,56]:(4)$$p_{ij}=\frac{p_{j|i}+p_{i|j\;}}{2N}$$

Points within the embedding space are grouped according to their pairwise conditional probability [55,56]:(5)$$q_{ij\;}=\frac{{\left(1+\|y_{i}-y_{j}{\|}^{2}\right)}^{-1}}{\sum_{k\neq i}{\left(1+\|y_{k}-y_{l}{\|}^{2}\right)}^{-1}}$$

This highlights the main function of t-SNE, which is to arrange the *n* points within a dataset in a low-dimensional space such that the difference between $q_{ij}$ and $p_{ij}$ is minimized using the Kullback–Leibler divergence.

t-SNE decomposition has previously been used to cluster and visualise Raman spectra from single neural pluripotent stem cells at various stages of differentiation and cell lines of various lineages exposed to various different chemotherapeutic agents [57,58]. As an approach to the decomposition of large multi-variate datasets t-SNE preserves local data structures at the expense of global structure, in comparison with PCA which attempts to preserve global data structure [56]. However, as t-SNE is an iterative process it cannot be employed to decompose a dataset into a stable set of reduced features for application to held out testing datasets within supervised machine learning.

Kobak et al. have investigated approaches to allow the preservation of both global and local structure using t-SNE with transcriptomics data [56]. To obtain meaningful visualisation of data t-SNE has various hyperparameters which must be tuned, including learning rate, perplexity, exaggeration and number of iterations. The number of iterations is generally set such that the visualisation evolves to a stable configuration. The learning rate on the other hand is recommended by Kobak et al. [56] to be set to the number of data points divided by 12. Care must be taken here to take account of the lack of harmonisation between t-SNE learning rates within different algorithms as they can differ by as much as a factor of 4. Kobak et al. also recommend setting large perplexity values of ~1% of the total number of data points for smaller datasets or a default value of 30 for smaller datasets. Their final recommendation is to use an exaggeration factor of 4 to allow separation of scores of classes into compact groups. In the present work we have tuned this parameter to provide a balance between cluster size and separation.

##### 4.2.4. Extreme-Gradient-Boosted Regression (XGBR)

XGBR is a form of gradient boosted decision tree regression which provides the means to improve model performance and execution time through implementing a gradient descent approach to optimise model complexity (number of decision tree leaves and branches) together with boosting to provide a consensus, low uncertainty prediction of regression target across multiple models [59]. While the algorithm has received significant attention in the computational literature it has received little attention in this regard in the bio-spectroscopy literature, where traditional chemometric regression approaches such as partial least squares regression are preferred [60,61,62,63]. Here, XGBR is used to perform regression of the PNT1A cell line spectra against data on the cell volume measured as part of the reference work [5]. The original data is reproduced in Table 5 for reference purposes.

This regression approach has the added value of allowing the identification of spectral regions of interest which maximise the regression performance, or in this case, variables which are associated with cell survival post exposure within the context of the irradiation approach for the PNT1A cell lines as described earlier.

For the optimisation of the XGBoost models the data was split into a training set and test set in a 70:30 split. XGBoost regression models hyperparameters were optimised via grid search cross-validation within scikit-learn over 20 separate epochs of data randomisation and those hyperparameters providing a minimised root mean squared error of prediction (RMSEP) on the held out test set were retained. The best set of spectral features was also output together with the frequency with which they were selected across all randomisation epochs.

##### 4.2.5. Classic Least Squares Spectral Fitting (CLS)

Classic least squares fitting is a powerful approach to the decomposition of spectra into components and fit coefficients for known constituents of chemical mixtures or samples. Previously we have used this approach for the elucidation of the relative changes in concentrations of biochemical species within cells and liquid biopsies in the context of identifying molecular markers of radiation induced toxicity. Briefly the approach decomposes a matrix of spectra, *X*, into the concentration, *c*, of a set of constituent molecular spectra, *R*, with *ε* representing a matrix of fitting residuals:(6)X=c.R+ε

In the present work this approach to spectral decomposition was used for the purposes of elucidation of the main spectral variants post exposure to ionising radiation and secreted factors in the context of the experiment. Within the fitting procedure all fitting coefficients were constrained to positive values.

For this exercise a total of 31 reference spectra were utilised, which included spectra of nucleic acids (DNA, RNA), protein (actin, keratin, ubiquitin, histone), glycoprotein (apolipiprotein-E3, apoliprotein-E4), lipid (phosphatidylcholine, phosphatidylinositol, phosphatidylethanolamine, phosphatidylserine, ceramide), carbohydrate (glycogen), nucleoside (ATP), cytokines (IL1, IL6, IL8), antioxidants (catalase, cysteine, glutathione (in both oxidised and reduced forms), vitamin C, vitamin E, tryptophan, β-carotene) and various signalling molecules of radiobiological importance (cytochrome-C, TGF-β1, TGF-β2, TNF-α, protein-kinase K). For all spectra processing was conducted in OCTAVVS before CLS regression [45]. All reference molecules were purchased from Sigma-Aldrich and their spectra were acquired as described elsewhere [10].

## Figures and Tables

**Figure 1 ijms-23-12986-f001:**
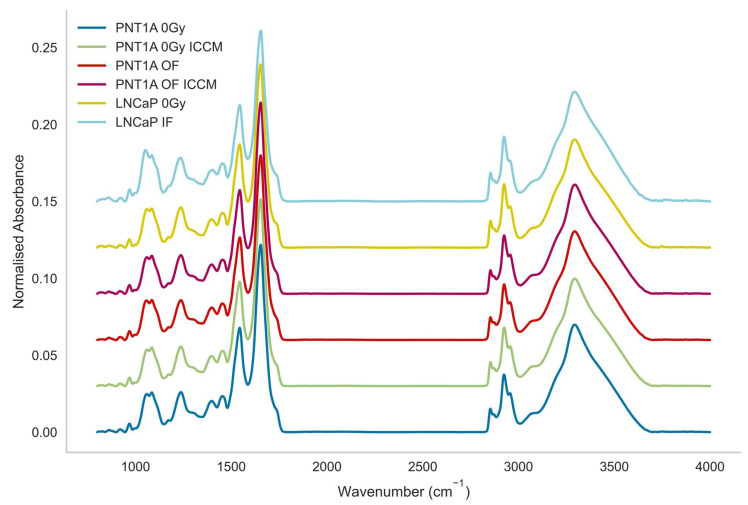
Mean spectra of all cell lines by class.

**Figure 2 ijms-23-12986-f002:**
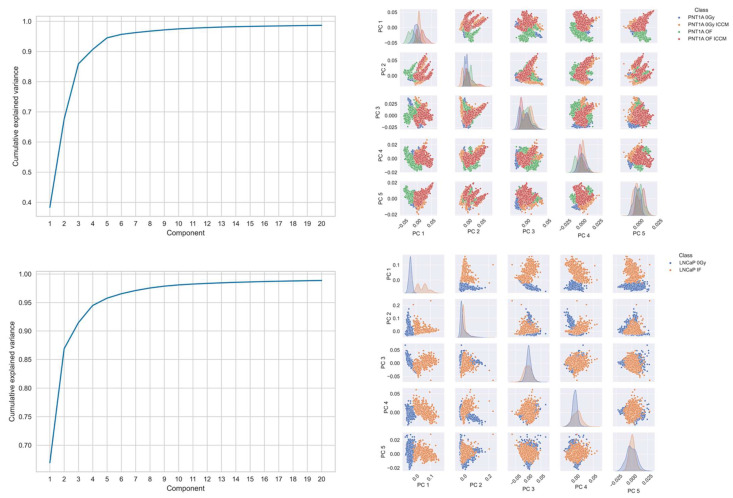
Principal components analysis of (**top** panel) PNT1A cell line and (**bottom** panel) LNCaP cell line spectra. In each panel the left figure displays the cumulative explained variance and the right displays a pairs plot of the principal component scores for the first 5 principal components.

**Figure 3 ijms-23-12986-f003:**
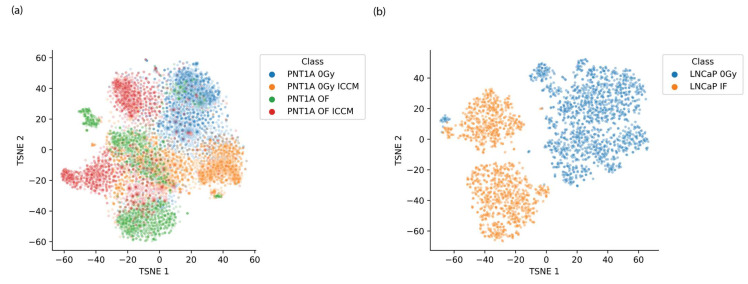
Representative t-SNE scores plots for (**a**) PNT1A cells exposed to a range of treatment modes and (**b**) LNCaP cells exposed to 2 Gy (IF) and 0 Gy (Control).

**Figure 4 ijms-23-12986-f004:**
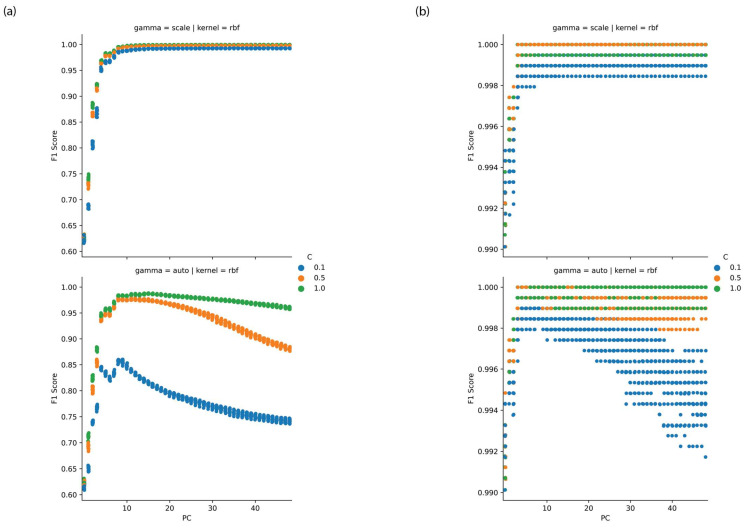
Performance of PCA-SVM versus principal component for (**a**) PNT1A and (**b**) LNCaP cell lines with variation in SVM hyperparameters.

**Figure 5 ijms-23-12986-f005:**
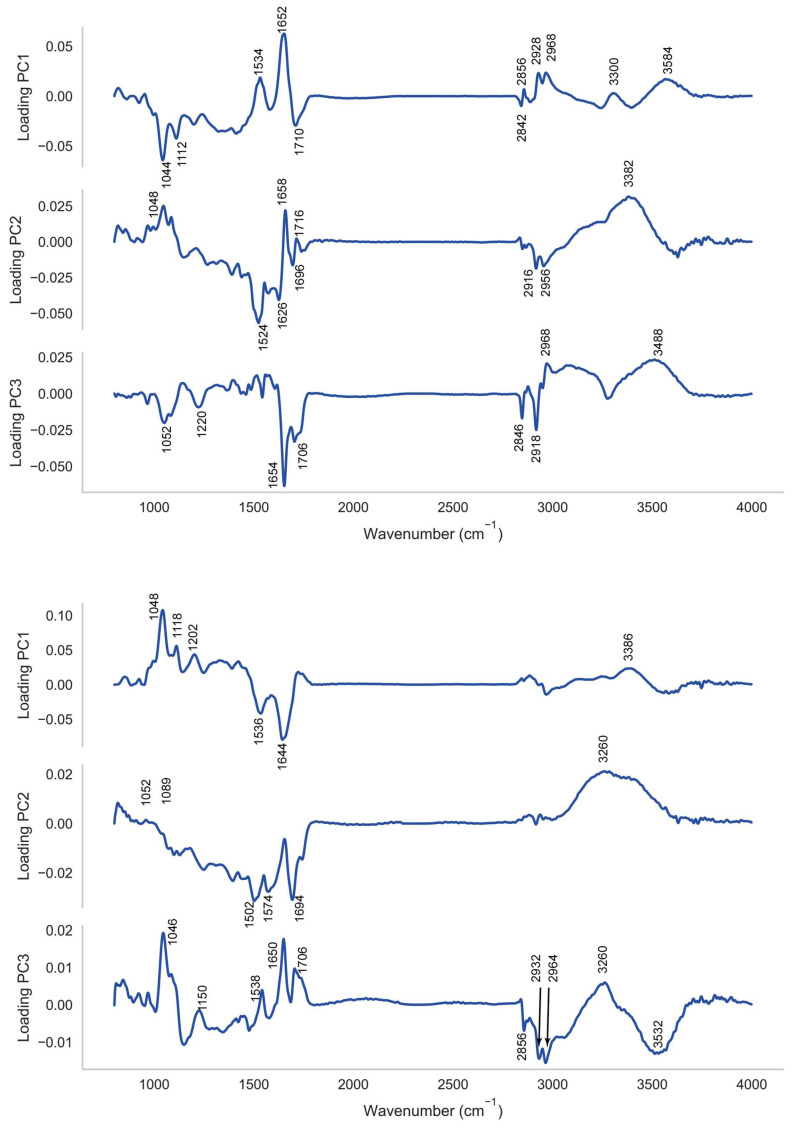
Loadings to the first three principal components for (**top**) analysis of PNT1A and (**bottom**) LNCaP PCA. Specific loadings of interest are highlighted.

**Figure 6 ijms-23-12986-f006:**
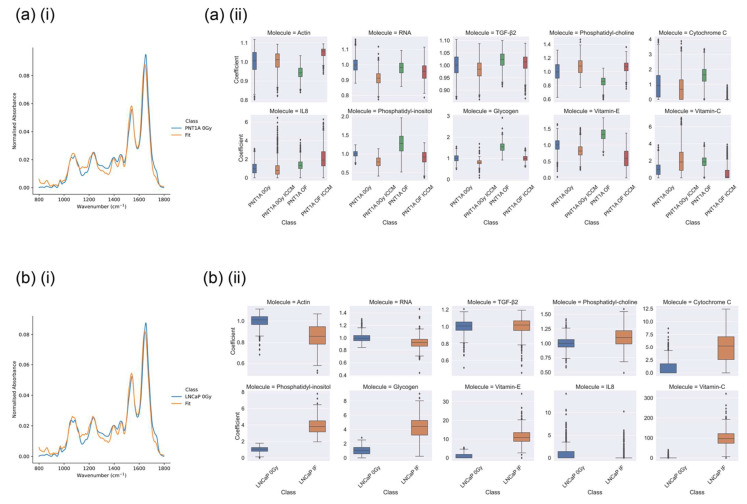
Examples of (**i**) fitting of mean spectra and (**ii**) CLS regression coefficients for (**a**) PNT1A and (**b**) LNCaP cell lines in which non-zero regression coefficients were observed (**i**) PNT1A and (**ii**) LNCaP cell lines in the fingerprint region using CLS regression. All regression coefficients have been normalised to the sham-irradiated (0 Gy) control for each cell line.

**Figure 7 ijms-23-12986-f007:**
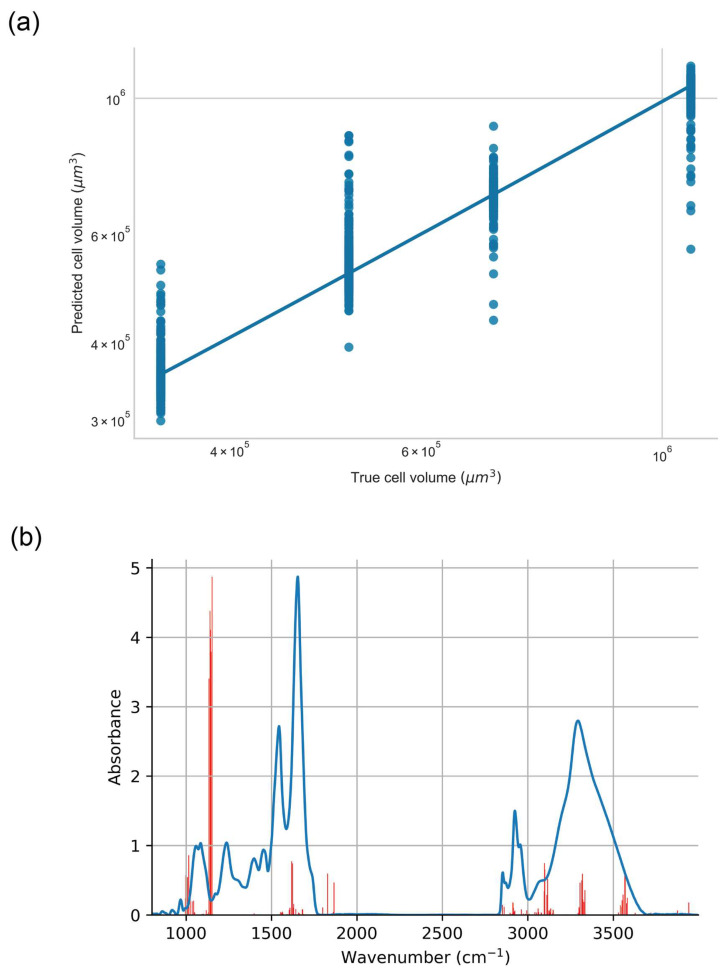
XGBoost regression of IR spectra from PNT1A cell line against cell volume. (**a**) Association between spectral data and cell volume. (**b**) Frequency with which spectral features (highlighted in red) are selected by XGBoost model in regression against cell volume.

**Table 1 ijms-23-12986-t001:** Results of testing via Welch’s ANOVA on CLS coefficients from fitting of PNT1A cell spectra.

F	*p*-Value	Molecule
3246	<0.001	Actin
3836	<0.001	Cytochrome C
4143	<0.001	Glycogen
418	<0.001	IL8
2352	<0.001	Phosphatidyl-choline
1531	<0.001	Phosphatidyl-inositol
1006	<0.001	RNA
336	<0.001	TGF-β2
1092	<0.001	Vitamin-C
2657	<0.001	Vitamin-E

**Table 2 ijms-23-12986-t002:** Results of post hoc pairwise testing with the Games-Howell approach on CLS coefficients from fitting of PNT1A cell spectra. Column A and B of the table identify the paired classes which are compared for each molecule, with SE denoting the standard error on the mean and T the t-statistic. Non-significant *p*-values at the level of *p* < 0.001 are highlighted in red.

A	B	SE	T	*p*-Value	Molecule
PNT1A 0 Gy	PNT1A 0 Gy ICCM	1.964 × 10^−3^	−2.4	0.074	Actin
PNT1A 0 Gy	PNT1A OF	1.713 × 10^−3^	32.4	<0.001	Actin
PNT1A 0 Gy	PNT1A OF ICCM	1.630 × 10^−3^	−30.2	<0.001	Actin
PNT1A 0 Gy	PNT1A 0 Gy ICCM	3.004 × 10^−2^	6.4	<0.001	Cytochrome C
PNT1A 0 Gy	PNT1A OF	2.772 × 10^−2^	−20.9	<0.001	Cytochrome C
PNT1A 0 Gy	PNT1A OF ICCM	2.255 × 10^−2^	42.9	<0.001	Cytochrome C
PNT1A 0 Gy	PNT1A 0 Gy ICCM	5.284 × 10^−3^	35.9	<0.001	Glycogen
PNT1A 0 Gy	PNT1A OF	7.196 × 10^−3^	−73.6	<0.001	Glycogen
PNT1A 0 Gy	PNT1A OF ICCM	5.642 × 10^−3^	1.0	0.757	Glycogen
PNT1A 0 Gy	PNT1A 0 Gy ICCM	2.950 × 10^−2^	−0.2	0.997	IL8
PNT1A 0 Gy	PNT1A OF	2.188 × 10^−2^	−16.9	<0.001	IL8
PNT1A 0 Gy	PNT1A OF ICCM	3.463 × 10^−2^	−33.0	<0.001	IL8
PNT1A 0 Gy	PNT1A 0 Gy ICCM	4.661 × 10^−3^	−16.7	<0.001	Phosphatidyl-choline
PNT1A 0 Gy	PNT1A OF	3.810 × 10^−3^	37.9	<0.001	Phosphatidyl-choline
PNT1A 0 Gy	PNT1A OF ICCM	3.966 × 10^−3^	−17.7	<0.001	Phosphatidyl-choline
PNT1A 0 Gy	PNT1A 0 Gy ICCM	4.236 × 10^−3^	51.3	<0.001	Phosphatidyl-inositol
PNT1A 0 Gy	PNT1A OF	7.814 × 10^−3^	−34.7	<0.001	Phosphatidyl-inositol
PNT1A 0 Gy	PNT1A OF ICCM	5.190 × 10^−3^	18.6	<0.001	Phosphatidyl-inositol
PNT1A 0 Gy	PNT1A 0 Gy ICCM	1.706 × 10^−3^	51.5	<0.001	RNA
PNT1A 0 Gy	PNT1A OF	1.721 × 10^−3^	12.1	<0.001	RNA
PNT1A 0 Gy	PNT1A OF ICCM	1.933 × 10^−3^	23.5	<0.001	RNA
PNT1A 0 Gy	PNT1A 0 Gy ICCM	1.507 × 10^−3^	11.5	<0.001	TGF-β2
PNT1A 0 Gy	PNT1A OF	1.429 × 10^−3^	−15.3	<0.001	TGF-β2
PNT1A 0 Gy	PNT1A OF ICCM	1.486 × 10^−3^	−5.6	<0.001	TGF-β2
PNT1A 0 Gy	PNT1A 0 Gy ICCM	4.315 × 10^−2^	−23.6	<0.001	Vitamin-C
PNT1A 0 Gy	PNT1A OF	2.655 × 10^−2^	−33.8	<0.001	Vitamin-C
PNT1A 0 Gy	PNT1A OF ICCM	2.770 × 10^−2^	17.5	<0.001	Vitamin-C
PNT1A 0 Gy	PNT1A 0 Gy ICCM	7.249 × 10^−3^	23.2	<0.001	Vitamin-E
PNT1A 0 Gy	PNT1A OF	7.169 × 10^−3^	−45.8	<0.001	Vitamin-E
PNT1A 0 Gy	PNT1A OF ICCM	9.688 × 10^−3^	43.5	<0.001	Vitamin-E

**Table 3 ijms-23-12986-t003:** Results of testing via Welch’s ANOVA on CLS coefficients from fitting of LNCaP cell spectra.

F	*p*-Value	Molecule
1167	<0.001	Actin
1180	<0.001	Cytochrome C
3938	<0.001	Glycogen
106	<0.001	IL8
272	<0.001	Phosphatidyl-choline
7020	<0.001	Phosphatidyl-inositol
389	<0.001	RNA
1	<0.001	TGF-β2
5246	<0.001	Vitamin-C
5833	<0.001	Vitamin-E

**Table 4 ijms-23-12986-t004:** Results of post hoc pairwise testing with the Games-Howell approach on CLS coefficients from fitting of LNCaP cell spectra. Column A and B of the table identify the paired classes which are compared for each molecule, with SE denoting the standard error on the mean and T the t-statistic.

A	B	SE	T	*p*-Value	Molecule
LNCaP 0 Gy	LNCaP IF	0.004	34.2	<0.001	Actin
LNCaP 0 Gy	LNCaP IF	0.112	−34.4	<0.001	Cytochrome C
LNCaP 0 Gy	LNCaP IF	0.052	−62.8	<0.001	Glycogen
LNCaP 0 Gy	LNCaP IF	0.064	10.3	<0.001	IL8
LNCaP 0 Gy	LNCaP IF	0.006	−16.5	<0.001	Phosphatidyl-choline
LNCaP 0 Gy	LNCaP IF	0.034	−83.8	<0.001	Phosphatidyl-inositol
LNCaP 0 Gy	LNCaP IF	0.004	19.7	<0.001	RNA
LNCaP 0 Gy	LNCaP IF	0.004	−1.1	<0.001	TGF-β2
LNCaP 0 Gy	LNCaP IF	1.348	−72.4	<0.001	Vitamin-C
LNCaP 0 Gy	LNCaP IF	0.135	−76.4	<0.001	Vitamin-E

**Table 5 ijms-23-12986-t005:** Colony volume measurements of PNT1A cell cultures (n = 3) from Shields et al (unpublished data) [5].

Irradiation	Volume Average (μm^3^)	Std Dev
0 Gy	699,592	105,269
Out of field (OF)	345,629	192,621
0 Gy + ICCM	514,930	83,311
Out of field + ICCM (OF + ICCM)	1,062,630	254,959

**Table 6 ijms-23-12986-t006:** Summary of spectral changes observed by exposure mode.

Sample	Exposure Mode	Protein	Carbohydrate	RNA	Lipid	Cytokine	Antioxidants	Cytochrome C	DNA Damage	Cell Survival
PNT1A OF	Low dose	Decrease	Increase	No change	Increase	Increase	No change	Increase	Increase	Increase
PNT1A OF ICCM	Low dose plus exposure to secreted factors	Increase	No change	Decrease	No change	No change	Decrease	Decrease	Increase	Increase
PNTT1A ICCM	Exposure to secreted factors	No change	Decrease	Decrease	Increase	No change	Decrease	No change	No change	Decrease
LNCAP 2 Gy	Exposure to high doses	Decrease	Increase	Decrease	Increase	Decrease	Increase	Increase	Increase	Not measured

## Data Availability

Data is available from the corresponding author upon reasonable request.

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
