# Peer review of "Application of Advanced Non-Linear Spectral Decomposition and Regression Methods for Spectroscopic Analysis of Targeted and Non-Targeted Irradiation Effects in an In-Vitro Model"

_ijms, 2022, doi:10.3390/ijms232112986_

Round 1

Reviewer 1 Report

This paper reports the results of high dose irradiation, low-dose scatter irradiation and radiation- induced bystander exposure (RIBE) signaling each produce differential effects on the cell. The authors' results may represent an important contribution of radiation research. However, the present quality of the manuscript is insufficient for publication, please consider the following suggestions that might help to improve the manuscript.

Specific comments:

1Specific comments:

1.     In the research, please add more details, such as RIBE in prostate cancer, mechanism of RIBE, intercellular communication between cancer and normal cells and FTIR spectroscopy & RIBE in prostate cancer and normal tissue.

2.     Why used only 1 h for ICCM study? If any specific reason? It would be helpful if the authors give example or scenario to support its description.

3.     Please provide more information about figure 6 and table 4 in the manuscript.

4.     Table 3, The authors are requested to discuss more about the colony volume of PNT1A cells. Why the volume of  OF + ICCM is larger than control/ OF or ICCM?

Author Response

We thank the reviewer for their detailed reading of the manuscript.

In responding to the first comment we have included a section outlining the bystander response and its importance to cancer therapy, and a section reviewing the applications of FTIR spectroscopy for the analysis of bystander effects within the introduction on pages 2 and 3 of the manuscript. This section is highlighted in yellow.

In responding to comment 2, ICCM was harvested from irradiated LNCaP cells one hour after irradiation as this is standard procedure in our laboratory for preparation of ICCM. Early work by Mothersill and Seymour (1997) investigated the effect of time post irradiation of medium transfer and showed no significant difference in clonogenic survival 1 - 60 hours post irradiation. A statement to this effect has been included within the manuscript on page 11. This section is highlighted in yellow.

In responding to comment 3, we have firstly noted an error with the construction of Table 4, whereby the labelling of rows 1 and 2 were exchanged inadvertently during formatting leading to a disconnect between the table, the text of the discussion and figure 6. We apologise for this error and have corrected it in the revised submission. Together with this change we have also added three extra columns to Table 4 to provide a greater degree of clarity regarding the connection between the CLS results in figure 6, the text of the discussion and the overall picture which emerges regarding the association between molecular events observed spectrally and either DNA damage or cell fate. These changes are throughout Table 4. We have also included explicit references to Table 4 throughout the discussion section and have included a new summary commentary at the end of the discussion. All of these  changes are highlighted in yellow in the manuscript.

As per the conclusions of Shields et al [5] we suspect that the increase in colony volume (and increase in cell survival observed by Shields) in the PNT1A OF ICCM cells is a signature of the adaptive response, whereby PNT1A cells have an increased protective response to exposure to RIBE factors through a priming out-of-field dose, a feature which has been observed in previous work [34]. We have included a statement to this effect in the Discussion section, paragraph 3, on page 9 of the manuscript. This section is highlighted in yellow.

Reviewer 2 Report

Dear Authors,

the paper "Application of advanced non-linear spectral decomposition and regression methods for spectroscopic analysis of targeted and non-targeted irradiation effects in an in-vitro model" by Slattery et al.  is a rather interesting study that opens up new opportunities for improving tumor response in terms of OS and PFS. Prostate cancer is the fourth most common cancer and the second most commonly diagnosed cancer in men worldwide. Three-dimensional radiation therapy has now been replaced by dynamic techniques such as intensity-modulated radiation therapy (IMRT) and volumetric modulated arc therapy (VMAT). A major concern in the use of VMAT is the possible increase of low-dose radiation to surrounding normal tissues, which could increase the risk of complications such as rectal bleeding and secondary malignancies. Interest in out-of-field radiation dose has increased with the introduction of these new techniques that offer superior compliance of high-dose regions to the target compared to conventional techniques, however with VMAT more normal tissues are exposed to low-dose radiation. There is a potential increase in radiobiological efficacy associated with lower energy photons delivered during VMAT, as normal cells are exposed to a temporal variation in the energy spectrum of incident photons. During VMAT, normal cells may be exposed to the primary radiation beam as well as transmission and scattering radiation. The impact of low-dose radiation, radiation-induced bystander effect and energy spectrum variation on normal cells is not well understood. This concern greatly affects late side effects, especially because of the longer life expectancy of prostate cancer patients. The identification of genetic and epigenetic markers that can differentially monitor the response of cells to radiotherapy treatments is highly valued. The study design is well described and the results support the preliminary hypotheses.

Spectroscopic techniques, such as FT-IR imaging spectroscopy, are a potentially attractive bioimaging platform that can provide molecular imaging. This technique offers high resolution and provides detailed information on biochemical and chemical features at the cellular and/or subcellular level, enabling the identification of pathology-related markers. Unlike standard histologic staining methods, FTIR imaging has the ability to simultaneously detect discrete changes in tissue structure and molecular composition. Direct biochemical analyses of all macromolecular components within tissue samples can be obtained from a single data acquisition, without the addition of dyes or chemical reagents and without altering tissue morphology). Infrared (IR) spectra are dominated by macromolecular building blocks, such as proteins, lipids, cholesterols, phospholipids, carbohydrates, and nucleic acids. IR spectral absorptions provide insights into these biomolecules and produce a sample-specific biochemical fingerprint. It is therefore a very powerful tool. In addition, the authors have applied advanced nonlinear spectral decomposition and regression methods for spectroscopic analysis. The whole makes the paper almost complete. The markers identified were able to discriminate between different exposure conditions and distinguish direct radiation exposure from secondary effects that may be responsible for cellular damage outside the irradiation field.

A limitation of this work is the 2D model chosen for the experiments. Cell cultures are a far away from the complex tissue and organ systems treated by Radiation Therapy, and the transferability of these interesting results to the clinic is currently not yet applicable. Cell and Organ chips together or as substitution of animal model are now widely used and allow a more specific tool for the transferability of basic scientific results to the clinics.

The authors are requested to briefly discuss these two aspects in the discussion and conclusions as future research and development.

Author Response

We thank this reviewer for their strong support of our work. To address their comment we have included a statement in the conclusion of the manuscript highlighted in blue, which emphasizes the limitations of our 2D cell culture model and envisions future work to overcome this limitation. This change has been highlighted in blue in the manuscript